# Epidemiology and Clinical Presentation of Children Hospitalized with SARS-CoV-2 Infection in Suburbs of Paris

**DOI:** 10.3390/jcm9072227

**Published:** 2020-07-14

**Authors:** Louise Gaborieau, Celine Delestrain, Philippe Bensaid, Audrey Vizeneux, Philippe Blanc, Aurélie Garraffo, Emilie Georget, Arnaud Chalvon, Nathalie Garrec, Yacine Laoudi, Emmanuelle Varon, Sébastien Rouget, Alexandre Pupin, Khaled Abdel Aal, David Toulorge, Sarah Ducrocq, Catherine Barrey, Letitia Pantalone, Blandine Robert, Lydie Joly-Sanchez, Caroline Thach, Caroline Masserot-Lureau, Jamilé Chahine, Veronica Risso Garcia-Roudaut, Jonathan Rozental, Sylvie Nathanson, Mohamed Khaled, Alexis Mandelcwajg, Nadia Demayer, Stéphanie Muller, Mustapha Mazerghane, Ralph Epaud, Béatrice Pellegrino, Fouad Madhi

**Affiliations:** 1Service de Pédiatrie Générale, Centre Hospitalier Intercommunal de Créteil, 94000 Créteil, France; louisegaborieau@gmail.com (L.G.); celine.delestrain@chicreteil.fr (C.D.); ralph.epaud@chicreteil.fr (R.E.); 2UPMC, Sorbonne université, 75006 Paris, France; 3INSERM, U955, Team GEIC2O, 94000 Créteil, France; 4INSERM, IMRB, Univ Paris Est Creteil, F-94010 Creteil, France; 5Centre des Maladies Respiratoires Rares (RESPIRARE^®^), 94000 Créteil, France; 6Service de Pédiatrie Générale, Centre Hospitalier Argenteuil, 95107 Argenteuil, France; philippe.bensaid@ch-argenteuil.fr (P.B.); audrey.vizeneux@ch-argenteuil.fr (A.V.); 7Service de pédiatrie Générale, Centre Hospitalier Intercommunal Poissy Saint-Germain en Laye, 78300 Poissy, France; philippe.blanc@ght-yvelinesnord.fr; 8Service de pédiatrie Générale, Centre Hospitalier Intercommunal Villeneuve Saint Georges, 94190 Villeneuve Saint George, France; aurelie.garraffo@chiv.fr (A.G.); emilie.georget@chiv.fr (E.G.); 9Site de Marne-la-Vallée, Service de pédiatrie Générale, Grand Hôpital de L’Est Francilien, 77600 Jossigny, France; achalvondemersay@ghef.fr (A.C.); ngarrec@ghef.fr (N.G.); 10Service de pédiatrie Générale, Centre Hospitalier Intercommunal Robert Ballanger, 93600 Aulnay-sous-Bois, France; yacine.laoudi@ght-gpne.fr; 11Laboratoire de Microbiologie, Centre Hospitalier Intercommunal de Créteil, 94000 Créteil, France; emmanuelle.varon@chicreteil.fr; 12Centre National de Référence des Pneumocoques, Centre Hospitalier Intercommunal Créteil, 94000 Créteil, France; 13Service de pédiatrie Générale, Centre Hospitalier Sud Francilien, 91100 Corbeil-Essonnes, France; sebastien.rouget@chsf.fr (S.R.); alexandre.pupin@chsf.fr (A.P.); 14Service de pédiatrie Générale, Centre Hospitalier Gonesse, 95500 Gonesse, France; abdel.aal@ch-gonesse.fr (K.A.A.); david.toulorge@ch-gonesse.fr (D.T.); 15Site Longjumeau, Service de pédiatrie Générale, Groupe Hospitalier Nord Essonne, 91160 Longjumeau, France; s.ducrocq@gh-nord-essonne.fr; 16Service de pédiatrie Générale, Hôpital Saint Camille, 94360 Bry-sur-Marne, France; c.barrey@ch-bry.org; 17Service de pédiatrie Générale, Centre Hospitalier René Dubos, 95300 Pontoise, France; letitia.pantalone@ght-novo.fr (L.P.); blandine.robert@ght-novo.fr (B.R.); 18Service de pédiatrie Générale, Centre Hospitalier de Melun, 77000 Melun, France; lydie.joly-sanchez@ch-melun.fr (L.J.-S.); caroline.thach@ch-melun.fr (C.T.); 19Site Meaux, Service de pédiatrie Générale, Grand Hôpital de L’Est Francilien, 77100 Meaux, France; cmasserot@ghef.fr; 20Service de pédiatrie Générale, Hôpital Simone Veil, 95600 Eaubonne, France; jamile.chahine@ch-simoneveil.fr; 21Service de pédiatrie Générale, Centre Hospitalier Fontainebleau, 77300 Fontainebleau, France; v.roudaut@ch-sud77.fr; 22Service de pédiatrie Générale, Hôpital Franco-Britannique, 92300 Levallois-Perret, France; jonathan.rozental@ihfb.org; 23Service de pédiatrie Générale, Centre Hospitalier de Versailles, 78150 Le Chesnay, France; snathanson@ch-versailles.fr; 24Service de pédiatrie Générale, Groupe Hospitalier Intercommunal Le Raincy Montfermeil, 93370 Montfermeil, France; mohamed.khaled@ght-gpne.fr; 25Service de pédiatrie Générale, Centre Hospitalier de Saint-Denis, 93200 Saint-Denis, France; alexis.mandelcwajg@ch-stdenis.fr; 26Service de pédiatrie Générale, Centre Hospitalier Arpajon, 91294 Arpajon, France; ldemayer@ch-arpajon.fr; 27Service de pédiatrie Générale, Centre Hospitalier de Rambouillet, 78120 Rambouillet, France; s.muller@ch-rambouillet.fr; 28Service de pédiatrie Générale, Centre Hospitalier André Grégoire, 93100 Montreuil, France; mustapha.mazeghrane@chi-andre-gregoire.fr; 29Service de pédiatrie Générale, Centre Hospitalier François Quesnay, 78200 Mantes-la-Jolie, France; b.pellegrino@ch-mantes.fr; 30IMRB-GRC GEMINI, Univ Paris Est Creteil, 94000 Créteil, France

**Keywords:** SARS-Co2 infection, clinical presentation, hospitalized children

## Abstract

Understanding the clinical presentation of severe acute respiratory syndrome coronavirus 2 (SARS-CoV-2) infection and prognosis in children is a major issue. Children often present mild symptoms, and some severe forms require paediatric intensive care, with in some cases a fatal prognosis. Our aim was to identify the epidemiological characteristics, clinical presentation, and prognosis of children with coronavirus disease 2019 (Covid-19) hospitalized in Paris suburb hospitals. In this prospective, observational, multicentre study, we included children hospitalized in paediatric departments of Paris suburb hospitals from 23 March 2020 to 10 May 2020, during the national lockdown in France with confirmed SARS-CoV-2 infection (positive RNA test on a nasopharyngeal swab) or highly suspected infection (clinical, biological, and/or radiological data features suggestive for SARS-CoV-2 infection). A total of 192 children were included for confirmed (*n* = 157) or highly suspected (*n* = 35) SARS-CoV-2 infection. The median age was one year old (interquartile range 0.125–11) with a sex ratio 1.3:1. Fever was recorded in 147 (76.6%) children and considered poorly tolerated in 29 (15.1%). The symptoms ranged from rhinorrhoea (34.4%) and gastrointestinal (35.5%) to respiratory distress (25%). Only 10 (5.2%) children had anosmia and five (2.6%) had chest pain. An underlying condition was identified in almost 30% of the children in our study. Overall, 24 (12.5%) children were admitted to paediatric intensive care units, 12 required mechanical ventilation, and three died. For children in Paris suburbs, most cases of Covid-19 showed mild or moderate clinical expression. However, one-eighth of children were admitted to paediatric intensive care units and three died.

## 1. Introduction

The discovery of a new form of pneumonia in early December 2019 in Wuhan, Hubei Province, followed by the rapid spread of the disease in China and across all continents, led to the most serious health crisis in the modern world for more than a century. Covid-19 results from the infection with the severe acute respiratory syndrome coronavirus-2 (SARS-CoV-2), a virus that is one of the six coronaviruses already known to infect humans. Four of these (HCoV-NL63, HCoV-229E, HCoV-OC43 and HKU1) usually caused mild common cold-type symptoms in immunocompetent people whereas the two others, the severe acute respiratory syndrome coronavirus (SARS-CoV) and the Middle East respiratory syndrome coronavirus (MERS-CoV) were responsible of severe pandemics in the past two decades [1]. 

SARS-CoV-2 is responsible for an ongoing global pandemic, leading to 4.2 million cases and 286,613 deaths as of 12 May 2020, with France among the main affected countries, with 26,646 deaths to date [2]. The first three Covid-19 cases identified in France were reported on 24 January 2020 in travellers returning from Wuhan, China [3]. On March 2020, the incidence of Covid-19 began to rapidly escalate in France. By March 23, France had the second highest number of Covid-19 infections and the greatest number of deaths in Europe, which led the French health authorities to initiate a lockdown from 17 March 2020.

Children are less commonly affected by SARS-CoV-2. The Chinese Centres for Disease Control and Prevention stated that, of the 72,314 cases reported as of 11 February 2020, only 1.3% were in individuals less than 19 years old [4]. Early reports from China, Italy, and the United States indicated that SARS-CoV-2 causes illness of varying degrees, with females and children underrepresented among cases, especially among severe and fatal cases [5,6,7]. Why children are less affected by SARS-CoV-2 is unclear and requires further investigation.

Children seem to have milder clinical symptoms than adults [5,6,7] (as has been reported for SARS-CoV and MERS-CoV infections) [8]. Nonetheless, asymptomatic or mildly symptomatic children might be at risk of transmitting the disease [9]. Most children infected with SARS-CoV-2 have been part of a family cluster outbreak. Also, some reports have shown that the contamination was more frequent in the adult–child direction and not the reverse, which suggests a lower transmission potential for children than adults [10,11]. 

Reports of infected children are growing but remain scarce [5,6,7] and only few clinical data are available for children with SARS-CoV-2 infection in France. The main objective of this study was to describe the clinical expression of Covid-19 in hospitalized paediatric patients in Paris suburbs between the start and end of house confinement established by the French health authorities.

## 2. Materials and Methods

### 2.1. Study Design and Patients 

We conducted a prospective observational multicentre study in 23 general paediatric hospitals located in Paris suburbs, one of the epicentres of the Covid-19 epidemic in France [12]. These 23 paediatric centres belong to the College of Hospital Paediatricians of Ile-de-France “COPHI” network (involving 23 general paediatrics departments, including paediatric emergency and neonatology departments, in Paris suburbs). With the national lockdown implemented in France on 17 March 2020, we conducted our study from 23 March 2020 (6 days after the lockdown) to 10 May 2020 (last day of lockdown), in a situation of reduced circulation of SARS-CoV-2 due to major social contact restrictions.

We included all children (< 18 years old) who were hospitalized with confirmed or highly suspected SARS-CoV-2 infection in one of the participating centres during the study period. We recorded demographic data, symptoms, clinical and biologic findings, imaging data, and outcomes. Non-hospitalized or untested patients were excluded from the analysis. 

The following clinical characteristics were considered: fever, upper respiratory-tract symptoms (cough, rhinitis, tonsillitis, odynophagia, otalgia, otitis, conjunctivitis), influenza-like illness (including asthenia, headache and myalgia), anosmia, dysgeusia, dyspnoea, chest pain, vomiting or diarrhoea, abdominal pain, skin involvement, arthritis or arthralgia, mucosal haemorrhage, Kawasaki syndrome (KD), and myocarditis. Covid-19 was suspected with any of these symptoms or signs as previously described [5,6,7,13,14,15]. 

### 2.2. SARS-CoV-2 RT-PCR Methods

Real-time polymerase chain reaction (RT-PCR) for SARS-CoV-2 was performed on a nasopharyngeal swab (NP) taken from each patient in the paediatric emergency department of each hospital at the time of hospitalization. The NP specimens were obtained by using the collection system eSwabTM (Minitip size nylon flocked swab placed in 1 mL of modified liquid Amies transport medium, COPAN, Brescia, Italy). Before extraction, each NP sample was inactivated by the addition of 750 µl/mL of STARmag lysis buffer solution (Seegne, South Korea). The RT-PCR for SARS-CoV-2 was performed on the automated Seegene STARlet system^®^, according to the manufacturer’s instructions using the CE marked AllplexTM 2019-nCoV RT-PCR assay (Seegene, South Korea^®^) which targets N- (viral nucleocapsid protein) and RdRP-gene (RNA-dependent RNApolymerase), both SARS-CoV-2 specific genes, and the sarbecovirus specific E-gene (viral envelop).

In brief, the automated Hamilton STARlet system was used for automated viral RNA extraction using the STARMag 96 Universal Cartridge kit (Seegene, South Korea) and PCR set up. Subsequently, 8 µl of extracted nucleic acids was added to 17 µL of the PCR Master Mix, and amplification and detection were performed on the CFW96TM detection system (Bio-Rad, France) as per manufacturer’s instruction. Ct from FAM (E gene), Cal Red 610 (RDEP gene), Quasar 670 (N gene) and HEX (internal control) were acquired. Before extraction, internal control (10 µl) was added to each reaction mix to verify extraction and determine PCR inhibition. Positive and negative controls were included in each run. NP samples were considered positive when a cycle threshold value (Ct) less than 40 was obtained for any gene. A sample was considered negative if the internal control was amplified, but not the viral target genes. A sample was considered invalid when no amplification was obtained for the internal control. The Ct values were used as indicators of the copy number of SARS-CoV-2 RNA in specimens with lower Ct values corresponding to higher viral copy numbers. 

### 2.3. Serological Assays 

Pediatricians collected fingerstick whole-blood specimens and used the Biosynex Covid-19 BSS test, a rapid chromatographic immunoassay, for qualitative detection of IgG and IgM antibodies to SARS-CoV-2 in blood. This test is among those approved by the French national health authority [16]. According to the specifications of the manufacturer, the diagnostic accuracy of the test was sensitivity 91.8% (95% CI 83.8–96.6) and specificity 99.2% (95% CI 97.7–99.8) (https://www.biosynex.com/laboratories-hopitaux-tests-covid-19/). Furthermore, assessment by independent investigators confirmed the good diagnostic accuracy of this test among hospital staff with mild disease in eastern France [17]. Positive serology was defined as a case positive for IgM and negative for IgG or positive for IgM and IgG or negative for IgM and positive for IgG. All other cases were considered to have negative serology results.

### 2.4. Definitions

Any patient with a positive RT-PCR result with a NP swab was considered to have confirmed SARS-CoV-2 infection. Highly suspected case was defined by a negative RT-PCR result for a NP with symptomatic children with one of the following criteria: anosmia or dysgeusia, household exposures with RT-PCR-confirmed or suspected cases, high-resolution computed tomography (HRCT) characteristics of Covid-19 (ground-glass opacities or bilateral lung consolidations, especially in the periphery), positive serology, and/or cardiac involvement (Kawasaki disease (KD) or myocarditis).

### 2.5. Ethics Approval

The study was approved by the institutional review board of the French society for respiratory medicine (Société de Pneumologie de Langue Française) (#CEPRO 2020-033).

### 2.6. Statistical Analysis

We describe case characteristics, including age, sex, history, co-existing conditions, exposure to SARS-CoV-2, symptoms, biological findings, and imaging patterns, when the case was reported. Chi-square and Fisher’s exact tests were used to compare categorical variables, and the Student *t* test was used for continuous variables. *p* < 0.05 was considered statistically significant. All analyses involved using GraphPad Prism 6.0^®^.

## 3. Results

Among the 205 children who were admitted to paediatric emergency departments in 23 Paris suburb hospitals during the study period, 192 (93.7%) were hospitalized and had available clinical data. The median age was 1 year (range 0.125–11), with a sex ratio of 1.3:1. RT-PCR results were negative for 35 children (18.2%) and were associated with age greater than one year, immunocompromised condition and nonsteroidal anti-inflammatory drugs (NSAIDs) (Table 1). The symptoms such as anosmia, dysgeusia and chest pain have been observed in children over six years of age. Table 1 summarizes the demographic and epidemiologic characteristics of the study population.

The underlying conditions were reported for 56 (29.2%) children, including mostly sickle cell disease (*n* = 16, 8.3%), asthma (*n* = 10, 5.2%), immunocompromised condition (*n* = 9, 4.7%), preterm birth (*n* = 8, 4.2%) and obesity defined by BMI > 95% (*n* = 5, 2.6%). Among the eight others (4.2%), only one had diabetes and three had epileptic encephalopathy, one had mitochondrial cytopathy, one had Turner syndrome, and one had Crohn disease. More than half of our study population had been exposed to an index case or an adult with suspected SARS-CoV-2 infection. Finally, only three (1.6%) children had received NSAIDs, 1 of which was admitted to paediatric intensive care unit (PICU).

Of the 35 cases highly suspected Covid-19, 21 children had either anosmia or dysgeusia (*n* = 3), or positive HRCT (*n* = 14, one of whom also had anosmia or dysgeusia) or positive serology (*n* = 6, one of whom also had positive HRCT). Out of the remaining 14, all of them had clinical signs or symptoms and household exposure (except 1, the latter had a KD). Out of the total of 35, all had clinical signs or symptoms and household exposure affected 12 children. Among these ones, all had either a positive serology or a positive HRCT (except 1, the latter had a KD) (Table 2). 

In the highly suspected cases group, 11 patients had only household exposure as an additional criterion. Interestingly, six children who tested negative in RT-PCR, finally had positive serology, and four of them were hospitalized in PICU.

Most of the children had low levels of inflammatory biomarkers, with median C-reactive protein and procalcitonin levels of 5.7 mg/L (interquartile range (IQR) 1.75–48.8) and 0.175 ng/mL (IQR 0.1–0.44), respectively. C-reactive protein level was elevated only in children with bacterial co-infection or KD. Blood cell count remained normal in most cases, with median neutrophil count 4.3 × 10^9^/L (IQR 2.0–7.9) and median lymphocyte count 2.5 × 10^9^/L (IQR 1.4–4.8). A total of 20 (10.4%) children had a chest X-ray that suggested consolidations, and of them, three children showed a ground-glass pattern. Moreover, 85 (44.3%) children had normal chest X-ray results. HRCT was performed in 36 children and showed abnormalities in 26 (72.2%), including consolidations and/or ground-glass opacities.

Co-infection was identified in 16 (8.3%) children, including 10 with febrile urinary tract infection (UTI) with *E. coli*. Only one child had parvovirus B-19 co-infection and the others had bacterial co-infection with *Bordetella pertussis*, methicillin-susceptible *Staphylococcus aureus*, methicillin-resistant *Staphylococcus aureus*, *Proteus mirabilis,* or *Fusobacterium necrophorum* (*n* = 1 each).

Apart from Covid-19, the main diagnoses were pneumonia in 13 (6.8%) children, bronchiolitis and febrile UTI in 11 (5.7%), acute gastroenteritis in six (3.1%), and vaso-occlusive crisis in five (2.6%). A total of 14 children (7.3%) had KD and eight (4.2%) had myocarditis (two children had KD with myocarditis), including nine with KD and seven with myocarditis recorded in the last two weeks. Supplemental oxygen (nasal canula or high-concentration face mask) was used in 19 (9.9%) children.

Fourteen children with a positive RT-PCR were hospitalized for reasons other than a SARS-CoV-2 infection: accommodation for a social or asymptomatic reason (*n* = 7), head trauma (*n* = 2), diabetic ketoacidosis (*n* = 1), appendicitis (*n* = 1), burn injury (*n* = 1), attempted suicide (*n* =1), and foreign body inhalation (*n* = 1).

In all, 24 (12.5%) children were hospitalized in PICUs, with invasive ventilation required in 12 (6.3%). Of these children, 11 (45.8%) had underlying conditions including asthma (*n* = 3), sickle cell disease (*n* = 2), immunocompromised condition (*n* = 2), obesity (*n* = 2), epileptic encephalopathy (*n* = 1), and preterm birth (*n* = 1). Children hospitalized in PICUs with a negative RT-PCR result were twice as numerous as those with positive results (*p* = 0.05). The reasons for admission to PICU were: myocarditis (*n* = 8), KD (*n* = 7), sepsis (*n* = 2), status epilepticus (*n* = 1) and covid-19 related respiratory distress (*n* = 8). Two children had KD complicated by myocarditis. A five-year-old girl received extracorporeal membrane oxygenation following co-infection with varicella complicated by septic shock due to *S. aureus*. She finally died due to acute respiratory distress syndrome. This patient tested positive for SARS-CoV-2 on tracheal aspiration after two negative RT-PCR results. Two other 16-year-olds died (both without known comorbidities) from acute myocarditis for the first and *F. necrophorum* septicaemia for the second. Seven had KD (four with a positive RT-PCR result) and eight had myocarditis (five with a positive RT-PCR result).

## 4. Discussion

To the best of our knowledge, this is the first French prospective observational multicentre study of the epidemiological characteristics and clinical features of hospitalized paediatric cases of Covid-19. Among 192 paediatric cases of Covid-19 reported as of 10 May 2020, clinical presentations were mild, as described in other countries [5,6,7,18]. In fact, only a few of our children presented a severe form of Covid-19 requiring intensive care (12.5%), and three children died. In a recent meta-analysis, the percentage of admissions to the PICU was slightly higher by 2%, and mortality was estimated at 0.08% compared to 1.6% in our study. [19]. 

According to the CONFIDENCE study in Italy, the age group under 1 year was the most affected in our study population [4]. However, Chinese studies do not report this high frequency in this age group [6,20]. Slightly more boys than girls (57.3% vs. 42.7%) were affected in our cohort, which is similar to previous study [5,6,7].

About 30% of the children in our cohort had an underlying condition, which is consistent with the Italian and US series which report that almost one-third of their patients had one. [5,6,7]. 

An interesting point from our study is the small number of immunocompromised children with SARS-CoV-2 infection. A similar observation was observed in this recent meta-analysis including 66 studies and 7480 children [21]. The other main underlying condition in our study was sickle cell disease, a disease prevalent in Paris suburbs.

Almost half of our children did not have intra-family transmission of SARS-CoV-2, as in the Italian CONFIDENCE cohort [5]. However, the Chinese and US paediatric series showed higher intra-familial exposure to SARS-CoV-2 [6,7]. Underdiagnosed asymptomatic cases are probably a feature of this viral infection and explain in part the difficulty in controlling the pandemic. We have little evidence regarding the transmission of SARS-CoV-2 by children and many of the childhood cases are from family clusters, with children tending to be identified through contact tracing of adult cases [18]. This crucial point deserves further investigation.

Only a few patients were co-infected with another virus in this study, unlike other studies [22,23]. There are two hypotheses to explain this. The first is the time factor. These two previous studies were conducted in January and early February, when viral infections were still prevalent (especially influenza A and B virus infections). The first child in our study had a positive RT-PCR result on 23 March 2020, relatively late in the viral epidemic. The second hypothesis is that we collected data during the lockdown established by the French authorities, which may explain a decrease in virus spread in the population and therefore less contamination. The Chinese articles were written just before the confinement in China [23] and two weeks later (the confinement in China began on 23 January 2020, with a reinforcement of the home confinement on 17 February by a total ban on all exits) [22]. Co-infections, especially viral, may be more numerous in the southern hemisphere, where the Covid-19 epidemic coincides with influenza and acute bronchiolitis epidemics. 

Children are reported to have milder SARS-CoV-2 infection, but some will show a severe form requiring admission to a PICU. In our study, 12.5% of children were admitted to a PICU which is higher than what is indicated in the WHO-China Joint Mission report on Covid-19 (2.5% severe cases in children < 19 years and only 0.2% critical cases) [24].

At the time of writing this article, some children were in a PICU for KD and myocarditis with inflammatory parameters. A press release from the paediatric societies dated 29 April 2020, warned of a resurgence of KD and myocarditis in an inflammatory context, possibly related to Covid-19. We decided to include these children, even in the absence of positive RT-PCR results because cases were likely an inflammatory reaction following infection due to SARS-CoV-2 in the previous weeks. Several of these children tested negative in RT-PCR for SARS-CoV-2 but some of them had a positive serology. Thus, these diseases could be a post-infectious consequence of the virus. Because they appeared in the last two weeks of the study is a further argument. A French national study is currently investigating these patients to understand the vascular tropism of SARS-CoV-2 and the pathophysiological mechanisms of the vasculitis in these children.

The presence of negatives results by RT-PCR is mainly linked to a test carried out late in the course of the disease. Likewise, the sensitivity is excellent (> 90%) and the specificity is 100% if the RT-PCR is carried out within 5 days of the onset of symptoms [25]. Serology will probably be of great help to allow screening of patients and their exposure at home if they are tested after the first week of the onset of symptoms. One study of 164 contact cases found that 16 patients with a positive RT-PCR result were all seropositive and seven patients with negative RT-PCR results had positive serology [26]. In our study, among the seven patients with positive serology, six had negative results on RT-PCR. In addition, a child was tested twice and was negative with NP samples, but the results of RT-PCR were ultimately positive while the sample was endobronchial. This observation, together with others, motivated our decision to include patients with negative RT-PCR results and strong suspicion of Covid-19 based on clinical and/or radiological findings.

Our study has several strengths. First, this is the first prospective multicentre study, to date, reporting the epidemiological and clinical characteristics of Covid-19 in children in Paris suburbs, which gave us a fairly broad view of the children hospitalized for this new infectious disease. Second, each centre declared and kept up to date a register for hospitalized children based on suspected or confirmed SARS-CoV-2 infection by RT-PCR performed by their laboratories during the inclusion period. Third, it is a homogeneous cohort of children with a confirmed or suspected SARS-CoV-2 infection who are symptomatic enough to be hospitalized.

This study also has some limitations. First, given that the available tests were limited in France for Covid-19 at the time of our study, it is possible that our results are influenced by an ascertainment bias. Second, we analysed only hospitalized patients. We did not perform mass screening during the study period. Thus, some non-hospitalized patients with positive RT-PCR results may have escaped our collection.

For children in Paris suburbs, most cases of Covid-19 showed mild or moderate clinical expression. However, one-eighth of children were admitted to paediatric intensive care and three died. Increased vigilance and prolonged monitoring are necessary to detect serious Covid-19 forms in children. Concurrently, we observed a resurgence of a few cases of Kawasaki disease and myocarditis.

## 5. Conclusions

This multicentric French study confirms previous paediatrics studies in other countries that children with SARS-CoV-2 infection mostly show a mild form of Covid-19. SARS-CoV-2 infection usually affects children with no underlying condition, causing severe disease in rare cases and is associated with a low rate of death.

## Figures and Tables

**Table 1 jcm-09-02227-t001:** General characteristics of hospitalized children depending on positive or negative RT-PCR results for SARS-CoV-2.

	Total (*n* = 192)	Positive RT-PCR (*n* = 157)	Negative RT-PCR (*n* = 35)	*p* Value
Age, years	1 (0.125–11)	0.5 (0.125–10)	7.5 (2–12.5)	0.005
≤1 month	34 (17.7)	32 (20.4)	2 (5.7)	0.040
<1 year	94 (49)	90 (57.3)	4 (11.4)	ns
1–5 years	26 (13.5)	15 (9.6)	11 (31.4)	0.002
6–10 years	22 (11.5)	16 (10.2)	6 (17.1)	ns
11–18 years	50 (26)	36 (22.9)	14 (40)	0.037
Sex				
Male	110 (57.3)	94 (59.9)	16 (45.7)	ns
Female	82 (42.7)	63 (40.1)	19 (54.3)	
Underlying condition	56 (29.2)	44 (28)	12 (34.3)	ns
Sickle cell disease	16 (8.3)	14 (8.9)	2 (5.7)	ns
Asthma	10 (5.2)	7 (4.5)	3 (8.6)	ns
Immunocompromised condition	9 (4.7)	4 (2.5)	5 (14.3)	0.016
Preterm birth	8 (4.2)	7 (4.5)	1 (2.9)	ns
Obesity	5 (2.6)	4 (2.5)	1 (2.9)	ns
Others	8 (4.2)	8 (5.1)	0 (0)	ns
NSAIDs	3 (1.6)	0(0)	3 (8.6%)	0.006
Household exposure	109 (56.8)	86 (54.8)	23 (65.7)	ns
Symptoms/signs				
Fever	147 (76.6)	116 (73.9)	31 (88.6)	ns
Fever poorly tolerated	29 (15.1)	23 (14.6)	6 (17.1)	ns
Rhinorrhoea	66 (34.4)	57 (36.3)	9 (25.7)	ns
Deteriorated general condition	63 (32.8)	47 (29.9)	16 (45.7)	ns
Respiratory distress	48 (25)	38 (24.2)	10 (28.6)	ns
Diarrhoea	32 (16.7)	24 (15.3)	8 (22.9)	ns
Vomiting	19 (9.9)	12 (7.6)	7 (20)	ns
No feeding or feeding difficulty	17 (8.9)	16 (10.2)	1 (2.9)	ns
Anosmia, dysgeusia	10 (5.2)	7 (4.5)	3 (8.6)	ns
Chest pain	5 (2.6)	4 (2.5)	1 (2.9)	ns
Length of hospital stays, days	3 (2–4)	3 (2–4)	2 (2–5)	ns
PICU admission	24 (12.5)	16 (10.2)	8 (22.9)	0.050
Length of stay in PICU, days	6 (3–10)	7 (3.25–13.75)	5 (2.75–6.5)	ns
Respiratory support	19 (9.9)	14 (8.9)	5 (14.3)	ns
High-flow ventilation	3 (1.6)	1 (0.6)	2 (5.7)	ns
Non-invasive ventilation	4 (2.1)	4 (2.5)	0 (0)	ns
Mechanical ventilation	12 (6.3)	9 (5.7)	3 (8.6)	ns
Mortality	3 (1.6)	3 (1.9)	0 (0)	ns

Quantitative data are presented as median (Q1–Q3) and categorical data as number (%). NSAIDs: nonsteroidal anti-inflammatory drugs; PICU: paediatric intensive care unit, ns: not significant.

**Table 2 jcm-09-02227-t002:** Description of the clinical and radiological profile of the 35 children with a negative RT-PCR in nasopharyngeal swab.

	Total, *n* = 35	Household Exposure *n* = 23	No Household Exposure, *n* = 12
Symptoms or clinical signs	35/35 (100)	23/23 (100)	12/12 (100)
Anosmia or dysgeusia	3/32 (9.3)	3 (13)	0 (0)
HRCT patterns	14/35 (40)	6 (26)	8/12 (66.7)
Serology	6/35 (17.1)	2 (8.7)	4/12 (33.3)
KD	9/35 (25.7)	4 (17.4)	5/12 (41.7)
Myocarditis	2/35 (5.7)	2 (8.7)	0/12 (0)

Categorical data are presented as number (%), HRCT: high resolution computed tomography, KD: Kawasaki Disease.

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
