# Peer review of "Epidemiology and Clinical Presentation of Children Hospitalized with SARS-CoV-2 Infection in Suburbs of Paris"

_jcm, 2020, doi:10.3390/jcm9072227_

Round 1

Reviewer 1 Report

Overall well written on prescription, however English language needs some minor improvements.

Specific remarks

Page 3 line 109: The definition of the highly suspected cases involves household exposure not only to prove with also suspected cases. Many include children only fulfilled this criterion?

It is not clear to me if you only tested for Covid in case of suspicious symptoms or if all children needing a hospital admission for example for urgent surgery were also screened? We had some positive children in the latter situation, for example admitted for an acute appendicitis that were positive on screening without specific covid symptoms. Please comment.

Table 1: Were there statistically different characteristics between positive and negative test of children? For example the median age should be. This seems to be related to the fact that the suspected/not proven cases involve more children with KD and myocarditis which occurs in older children. This is also reflected in the higher percentage of PICU admission in the suspected first proven cases.

Symptoms like Anosmia , dysgeusia and chest pain will only be reported by older children but may be present in younger children as well. Therefore it can be expected that these symptoms are more frequently present in older children. This should be mentioned.

How was obesity defined?

Line 211: I would argue that this child with varicella died with covid most likely not due to covid. It is disputable if this should be counted as a covid death. When screening all admissions in paediatric departments some children have a positive swab on screening while the main reason for admission is completely other disease which may be sometimes fatal.

The discussion is overall quite long compared to the rest of the manuscript with some topics being repeated. Also the logical structure of the discussion can be improved. In the current form the discussion is quite hard to read.

Line 217: The authors stated this is the first prospective observational multicentre study. They should mention in line 220 that previous studies are retrospective if that is the case.

Line 224: The reason that children below the age of 1 or more often admitted may be related to a general finding in paediatric infectious disease. In this age group the reason for admission with fever is often failure of feeding and need for IV fluid or NG feeding like is the case in other viral infections. Additionally in younger children Fever of unknown origin is also often reason for admission and further investigations. This may not be covered specific and not linked to the severity of the viral illness as such.

Line 234 and following: This paragraph should be rewritten. The message is not really clear to me. The sentence 236 starting with ‘indeed’: The message in this sentence form does not follow on the previous sentence and I am not sure what the authors mean with indeed.

Line 367: what do you mean by the last 2 weeks? Of registration for this study ? Of confinement?

Line 279 : please add that this was an endobrochial or bronchoscopic sample. This is a common finding in adults patients with covid infection suspected on chest CT.

Author Response

Dear Editor,

We thank you for giving us the opportunity to submit a revised version of our article entitled “Epidemiology and clinical presentation of children hospitalized with SARS-CoV-2 infection in suburbs of Paris”. We have taken into account your helpful comments to improve our work and provide a “point-by-point” response to the reviewer’s comments outlining the modifications made to the new manuscript.

RESPONSES TO REVIEWER 1

We thank Reviewer 1 for his constructive criticisms, and we have modified our manuscript accordingly.

Reviewer: 1

Page 3 line 109: The definition of the highly suspected cases involves household exposure not only to prove with also suspected cases. Many include children only fulfilled this criterion?

Response: You are absolutely right. The contact cases were not systematically tested by RT-PCR and therefore we could not know the exact number of confirmed cases in the family circle. Indeed, the French health authorities did not have the means in terms of tests to achieve this and therefore the recommendations during this period is to stay at home if the symptoms were minor and to consult only in the event of worsening of the symptoms.

To complete the answer to your question and minimize this bias, among the 23 highly suspected cases who had a household exposures, 12 patients had at least one of the following criteria: anosmia or dysgeusia (n=3), high-resolution computed tomography (HRCT) characteristics of Covid-19 (ground-glass opacities or bilateral lung consolidations, especially in the periphery) (n=6), positive serology (2) and/or cardiac involvement (Kawasaki disease (KD) (n=4) or myocarditis (n=2).

Only 11 patients had only household exposures. We added this information in the table 2 and in the results section (line 220-221 page 7).

It is not clear to me if you only tested for Covid in case of suspicious symptoms or if all children needing a hospital admission for example for urgent surgery were also screened? We had some positive children in the latter situation, for example admitted for an acute appendicitis that were positive on screening without specific covid symptoms. Please comment.

We thank the reviewer for this comment.

We included all children (<18 years old) who were hospitalized with confirmed or highly suspected SARS-CoV-2 infection in one of the participating centres during the study period. Real-time polymerase chain reaction (RT-PCR) for SARS-CoV-2 was performed on a nasopharyngeal swab (NP) taken from each patient in the paediatric emergency department of each hospital at the time of hospitalization.

Indeed, we included patients who were positive SARS-CoV-2 but hospitalized for a reason other than an infectious one because our screening policy at the time of the study was to perform a systemic RT-PCR on a NP in each hospitalized child.

14 children with a positive RT-PCR were hospitalized for a reason not in favor of a SARS-CoV-2 infection: accommodation for a social or asymptomatic reason (n=7), head trauma (n=2), diabetic ketoacidosis (n=1), appendicitis (n=1), burn injury (n=1), attempted suicide (n =1) and foreign body inhalation (n=1). We added this in the results section (line 240-243 page 7 and 8).

We have changed the order of the sentences in the methodology section for clarity.

Table 1: Were there statistically different characteristics between positive and negative test of children? For example the median age should be. This seems to be related to the fact that the suspected/not proven cases involve more children with KD and myocarditis which occurs in older children. This is also reflected in the higher percentage of PICU admission in the suspected first proven cases.

You are absolutely right. There is indeed a difference in terms of age and especially in the following ranges (<1 month, 1-5 and 11-18 y). Children between 1 and 5 years of age had more negative RT-PCR (p=0.0017). Older children (> 1 year) had more negative RT-PCR in our study group.This was added in the section results (line 160-161 page 4).

Symptoms like Anosmia , dysgeusia and chest pain will only be reported by older children but may be present in younger children as well. Therefore it can be expected that these symptoms are more frequently present in older children. This should be mentioned.

We agree. All children with anosmia and/or dysgeusia were 6 years of age or older. All children with chest pain were 13 years of age or older. We added this in the results section (line 161-162 page 4).

How was obesity defined?

If a child's BMI is higher than 95% (95 out of 100) of other children their age and sex, they are considered overweight or obese. We added this information in section results (line 174 page 6).

Line 211: I would argue that this child with varicella died with covid most likely not due to covid. It is disputable if this should be counted as a covid death. When screening all admissions in paediatric departments some children have a positive swab on screening while the main reason for admission is completely other disease which may be sometimes fatal.

You are absolutely right. We also had this feeling. It is difficult to attribute the death of this child to SARS-CoV-2 infection. The accountability of SARS-CoV-2 is assured only in cases of acute respiratory distress syndrome with multiple organ failure.

The discussion is overall quite long compared to the rest of the manuscript with some topics being repeated. Also the logical structure of the discussion can be improved. In the current form the discussion is quite hard to read.

We have shortened the discussion section. Some sentences have been suppressed.

Line 217: The authors stated this is the first prospective observational multicentre study. They should mention in line 220 that previous studies are retrospective if that is the case.

This is the first study in France that is prospective. The other studies have been carried out in other countries. We corrected this in the text by adding: “To the best of our knowledge, this is the first French prospective observational multicentre study of the epidemiological characteristics and clinical features of hospitalized paediatric cases of Covid-19. Among 192 paediatric cases of Covid-19 reported as of May 10, 2020, clinical presentations were mild, as described in previous studies in other countries [4,5,6,17]”.

Line 224: The reason that children below the age of 1 or more often admitted may be related to a general finding in paediatric infectious disease. In this age group the reason for admission with fever is often failure of feeding and need for IV fluid or NG feeding like is the case in other viral infections. Additionally in younger children Fever of unknown origin is also often reason for admission and further investigations. This may not be covered specific and not linked to the severity of the viral illness as such.

We agree with your comment.

Line 234 and following: This paragraph should be rewritten. The message is not really clear to me. The sentence 236 starting with ‘indeed’: The message in this sentence form does not follow on the previous sentence and I am not sure what the authors mean with indeed.

We have rewritten the paragraph and change the word indeed to "likewise ”.

Line 367: what do you mean by the last 2 weeks? Of registration for this study ? Of confinement?

Thank you for your comment. The majority of kawasaki disease appeared in the last 2 weeks of the study (9/14). We modify this phrase in the discussion section: “Because they appeared in the last 2 weeks of the study is another argument”.

Line 279 : please add that this was an endobrochial or bronchoscopic sample. This is a common finding in adults patients with covid infection suspected on chest CT.

We modified as requested.

“In addition, a child was tested twice and was negative with NP samples but the results of RT-PCR were ultimately positive while the sample was endobronchial.”

We thank you again for your time and consideration and we hope that the revised manuscript will be accepted for publication in JCM.

Sincerely yours

Fouad MADHI, MD

Reviewer 2 Report

In the paper titled “Epidemiology and clinical presentation of children hospitalized with SARS-CoV-2 infection in suburbs of Paris”, the authors described the clinical presentation of SARS-CoV-2 among children in 23 pediatric departments of Paris suburb hospitals. This paper is timely for another confirmation that there are less SARS-CoV-2 infection in children. However, for publication there are some things that needs clarification/modification:

  1. Line 44: The words one of in ‘we included children hospitalized in one of 23 pediatric departments of Paris’ should be omitted.
  2. Line 64: The comma in ‘March, 2020’ should be omitted.
  3. The section of SARS-CoV-2 RT-PCR methods should be rewritten. For example, the section talking about RT-PCR for SARS-CoV-2 in the first paragraph should be moved to second paragraph to ensure chronological process from specimen collection, RNA extraction, then RT-PCR.
  4. Line 116: spicemens should be specimens.
  5. Table 1: Why length of hospital stays are much shorter than the length of stay in PICU. Do the authors mean length of non-intensive care stays? Are PICU facilities located on different hospitals?
  6. N=12/35 in table 2 should be N=12.
  7. Line 191, 9 in 109 should be superscript. Also, 4.3x109 and 2.5x109 instead of 4.3 109 and 2.5 109.
  8. Line 271: The phrase ‘false negative’ should be rephrase to ‘negative results’ because false negative mean there is a confirmation that the negative result is false. The paper does not seek after this confirmation.
  9. Line 292: In ‘data did not include Paris paediatric centres’, do the authors mean central Paris to differentiate with Paris suburbs which is the location of studies?
  10. Line 293: Also, the same issue as number 9 in ‘PICU located in Paris’. Where do the authors mean with ‘The latter’?

Author Response

We thank Reviewer 2 for his constructive criticisms, and we have modified our manuscript accordingly.

  1. Line 44: The words one of in ‘we included children hospitalized in one of 23 pediatric departments of Paris’ should be omitted.

done

  1. Line 64: The comma in ‘March, 2020’ should be omitted.

done

  1. The section of SARS-CoV-2 RT-PCR methods should be rewritten. For example, the section talking about RT-PCR for SARS-CoV-2 in the first paragraph should be moved to second paragraph to ensure chronological process from specimen collection, RNA extraction, then RT-PCR.

We have rewritten the paragraph as requested

  1. Line 116: spicemens should be specimens.

Thanks for the correction

  1. Table 1: Why length of hospital stays are much shorter than the length of stay in PICU. Do the authors mean length of non-intensive care stays? Are PICU facilities located on different hospitals?

Exactly, it is a length of stay in conventional hospitalization in general pediatrics and not in intensive care which are located most of the time in different hospitals.

  1. N=12/35 in table 2 should be N=12.

Done

  1. Line 191, 9 in 109 should be superscript. Also, 4.3x109 and 2.5x109 instead of 4.3 109 and 2.5 109.

Thanks, we have modified.

  1. Line 271: The phrase ‘false negative’ should be rephrase to ‘negative results’ because false negative mean there is a confirmation that the negative result is false. The paper does not seek after this confirmation.

We agree and modified the manuscript accordingly

  1. Line 292: In ‘data did not include Paris paediatric centres’, do the authors mean central Paris to differentiate with Paris suburbs which is the location of studies?

There are several separate paediatric wards between the centre of Paris and the outskirts of Paris. Our observatory only concerned hospitals on the suburbs of Paris. We decided to remove this sentence from the discussion to avoid confusion.

  1. Line 293: Also, the same issue as number 9 in ‘PICU located in Paris’. Where do the authors mean with ‘The latter’?

Thank you for your comment.

We wanted to say that the pediatric intensive care units are all located in the center of Paris.

We decided to remove this sentence from the discussion to avoid confusion.

We thank you again for your time and consideration and we hope that the revised manuscript will be accepted for publication in JCM.

Sincerely yours

Fouad MADHI, MD
